
# The Influence of Assimilating Leaf Area Index in a Land Surface
# Model on Global Water Fluxes and Storages
Xinxuan Zhang[1], Viviana Maggioni[1], Azbina Rahman[1], Paul Houser[1], Yuan Xue[1], Timothy
Sauer[1], Sujay Kumar[2] and David Mocko[2]
1. George Mason University, Fairfax, VA, USA
2. Hydrological Sciences Laboratory, NASA Goddard Space Flight Center, Greenbelt, MD, USA
Submit to:
Hydrology and Earth System Sciences
September, 2019



**Abstract**
Vegetation plays a fundamental role not only in the energy and carbon cycle, but also the global
water balance by controlling surface evapotranspiration. Thus, accurately estimating vegetation-
related variables has the potential to improve our understanding and estimation of the dynamic
interactions between the water and carbon cycles. This study aims to assess to what extent a land
surface model can be optimized through the assimilation of leaf area index (LAI) observations at
the global scale. Two observing system simulation experiments (OSSEs) are performed to evaluate
the efficiency of assimilating LAI through an Ensemble Kalman Filter (EnKF) to estimate LAI,
evapotranspiration (ET), interception evaporation (CIE), canopy water storage (CWS), surface soil
moisture (SSM), and terrestrial water storage (TWS). Results show that the LAI data assimilation
framework effectively reduces errors in LAI simulations. LAI assimilation also improves the
model estimates of all the water flux and storage variables considered in this study (ET, CIE, CWS,
SSM, and TWS), even when the forcing precipitation is strongly positively biased (extremely wet
condition). However, it tends to worsen some of the model estimated water-related variables (SSM
and TWS) when the forcing precipitation is affected by a dry bias. This is attributed to the fact that
the amount of water in the land surface model is conservative and the LAI assimilation introduces
more vegetation, which requires more water than what available within the soil. Future work
should investigate a multi-variate data assimilation system that concurrently merges both LAI and
soil moisture (or TWS) observations.



## 1. Introduction

Terrestrial vegetation plays a vital role in the global water cycle, as it controls the surface evapotranspiration and the state of the carbon cycle. As shown in past literature, there exists a strong relationship between vegetation, precipitation, and soil moisture (Di et al., 1994; Farrar et al., 1994; Richard and Poccard, 1998; Adegoke and Carleton, 2002). Nevertheless, the role that vegetation and its dynamics play in the water cycle (for instance on the variability of precipitation) is extremely complex (Wang and Eltahir 2000; Wang et al. 2011). In the past half-century, these land surface processes and feedbacks have been examined through numerical modeling experiments (e.g., Kim and Wang 2007). In early generation land surface models (LSMs), the development stage of vegetation was prescribed by regularly updating vegetation variables, based on fixed lookup tables to simplify the model computation (Foley et al. 1996). This approach uses constant vegetation indices, e.g., the Leaf Area Index (LAI), throughout a certain period, while in reality the growth of vegetation continuously changes in response to weather and climate conditions. To overcome this deficiency, new generation LSMs are coupled with dynamic vegetation models that comprehensively simulate several biogeochemical processes (Woodward and Lomas 2004; Gibelin et al. 2006; Fisher et al. 2018). LSMs with a dynamic vegetation module are able to capture more detailed variations in plant productivity than traditional LAI methods (Kucharik et al. 2000; Arora 2002; Krinner et al. 2005).

LAI can also be estimated through observations from satellite sensors, such as the Moderate Resolution Imaging Spectroradiometer (MODIS, Pagano and Durham 1993; Justice et al. 2002), the Système Probatoire d'Observation de la Terre VEGETATION (SPOT-VGT, Baret et al. 2007), and the National Oceanic and Atmospheric Administration (NOAA) Advanced Very High Resolution Radiometer (AVHRR, Cracknell 1997). LAI products retrieved from different



satellite missions and sensors provide spatially and temporally varying LAI fields on a routine

basis at regional and global scales, including the Moderate Resolution Imaging Spectroradiometer

(MODIS) LAI (Myneni et al. 2002), the Global Land Surface Satellite (GLASS) LAI (Xiao et al.

2013), and the GLOBMAP LAI dataset (Liu et al. 2012), among others. Satellite-derived LAI

products were found to be affected by uncertainties due to the limitation of retrieval algorithms

and vegetation type sampling issues (Cohen and Justice 1999; Privette et al. 2002; Tian et al. 2002;

Morisette et al. 2002).

     A method to combine the inherently incorrect estimates from satellite observations and

model simulations is data assimilation (DA). One of the most common DA systems — the

Ensemble Kalman Filter (EnKF; Evensen 2003) — dynamically updates the model error

covariance information by producing an ensemble of model predictions, which are individual

model realizations perturbed by the assumed model error (Reichle et al. 2007). The ensemble

approach is widely used in hydrologic DA because of its flexibility with respect to the type of

model error (Crow and Wood 2003) and well suited to the nonlinear nature of land surface

processes (Reichle et al. 2002a, 2002b; Andreadis and Lettenmaier 2006; Durand and Margulis

2008; Kumar et al. 2008; Pan and Wood 2006; Pauwels and De Lannoy 2006; Zhou et al. 2006).

However, the use of an EnKF for the assimilation of LAI in LSMs has not been thoroughly

investigated in the past. Pauwels et al. (2007) proposed an observing system simulation experiment

(OSSE) to evaluate the performance of assimilating LAI in a hydrology-crop growth model by an

EnKF algorithm. Other studies have also tested simplified 1D-VAR and extended Kalman filter

methods for LAI assimilation (e.g., Sabater et al. 2008; Barbu et al. 2011; Fairbairn et al. 2017).

Recently, Kumar et al. (2019) assimilated GLASS LAI assimilation in a land surface model with

an EnKF across the Continental U.S. Some model simulated water budget terms were improved





through the assimilation procedure, especially in agricultural areas because the assimilation added
harvesting information to the model. Ling et al. (2019) assimilated LAI information at the global
scale with an Ensemble Adjust Kalman Filter (EAKF) algorithm and found that the assimilation is
more effective during the growing season. LAI assimilation also had positive impact on gross
primary production (GPP) and evapotranspiration (ET) in low latitude regions.
Nevertheless, most of the aforementioned studies mainly focused on the impact of LAI
assimilation on the model simulated LAI or vegetation biomass. Only a few studies discussed the
influences of LAI assimilation on the estimation of water variables such as soil moisture or
streamflow (Pauwels et al. 2007; Sabater et al. 2008) and most of them focused on small regions.
This work leverages upon these studies but aims to assess to what extent a land surface
model, especially the model estimations of water-related variables, can be optimized through the
assimilation of LAI observations at the global scale. As this study serves as a feasibility test to
quantify the impact of LAI assimilation on water cycle variables, an OSSE is chosen to investigate
the model's behavior. This guarantees that reference variables (often referred to as the "truth"),
which are synthetically produced, are available for quantifying the performance of the proposed
framework. Specifically, two OSSEs that apply an EnKF algorithm to an LSM model are
performed to evaluate the efficiency of assimilating LAI observations for estimating
evapotranspiration, interception evaporation, canopy water storage, surface soil moisture, and
terrestrial water storage.

**2. Methods and materials**
*2.1. Land surface model*



The Noah LSM with multi-parameterization options (Noah-MP 3.6, Niu et al. 2011; Yang et al.
2011) is adopted in this study. Noah-MP contains a separate vegetation canopy defined by a canopy
top and bottom, crown radius, and leaves with defined dimensions, orientation, density, and
radiometric properties (Niu et al. 2011). Multiple options are available for surface water infiltration,
runoff, groundwater transfer and storage including water table depth to an unconfined aquifer (Niu
et al. 2007), dynamic vegetation, canopy resistance, and frozen soil physics. Specifically, the
prognostic vegetation growth combines a Ball-Berry photosynthesis-based stomatal resistance
(Ball et al. 1987) with a dynamic vegetation model (Dickinson et al. 1998) which calculates the
carbon storages in various parts of the vegetation (leaf, stem, wood, and root) and the soil carbon
pools.
The Noah-MP 3.6 LSM has been implemented into the NASA Land Information System
(LIS; Peters-Lidard et al. 2007; Kumar et al. 2006). LIS is a software that provides an interagency
test bed for land surface modeling and data assimilation that allows customized systems to be built,
assembled and reconfigured easily, using shared plugins and standard interfaces. All the
experiments of Noah-MP in this study are setup through LIS. The Modern-Era Retrospective
analysis for Research and Applications Version 2 (MERRA-2; Gelaro et al. 2017) dataset served
as the meteorological forcings for Noah-MP. MERRA-2 is the latest atmospheric reanalysis
produced by the NASA Global Modeling and Assimilation Office (GMAO) and includes updates
from the Goddard Earth Observing System (GEOS). The meteorological forcing variables selected
from MERRA-2 include surface pressure, surface air temperature, surface specific humidity,
incident radiations, wind speed, and precipitation rate.
Five model output variables that describe terrestrial water fluxes and storages are
investigated in this work: Evapotranspiration (ET, defined as the sum of evaporation and the plant


transpiration [kg/m²s]), Canopy Interception Evaporation (CIE, defined as the evaporation of the
canopy intercepted water [kg/m²s]), Canopy Water Storage (CWS, defined as the amount of
canopy intercepted water in both liquid and ice phases [kg/m²]), Surface Soil Moisture (SSM,
defined as the water content in the top 10 cm of the soil column [m³/m³]), and Terrestrial Water
Storage (TWS, defined as the sum of all water storage on the land surface and in the subsurface
[mm]).

### 2.2. Experimental design

An OSSE is designed to understand the efficiency of assimilating LAI within Noah-MP version
3.6 using a one-dimensional EnKF algorithm (Reichle et al. 2010), when the precipitation forcing
data are strongly biased. Being the major driving force of the hydrological cycle, the quality of
input precipitation is critical for the accuracy of a land surface model. However, global
precipitation datasets are far from being perfect and often affected by large regional biases. For
example, the MERRA-2 precipitation dataset shows a widespread relative biases greater than 100%
in South Asia (Ghatak et al. 2018). Although an EnKF is optimal only under the assumption of
unbiasedness (which is not met in the proposed experimental setup), we want to investigate here
to what extent a LAI EnKF (even if sub-optimal) can improve water storages and fluxes under two
extreme conditions, i.e., a very dry and a very wet precipitation bias, knowing that such biases are
very plausible in the real world and often unknown (and therefore difficult to remove). The
proposed framework is evaluated on a global scale (Antarctica excluded) at the 0.625° × 0.5°
spatial resolution of the MERRA-2 forcing dataset (Figure 1).



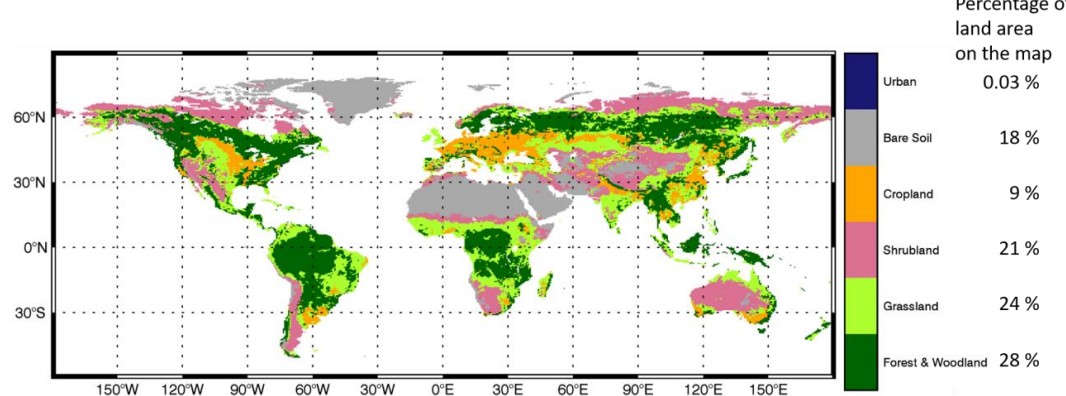


Figure 1. Study domain and land cover types (Hansen et al. 2000).

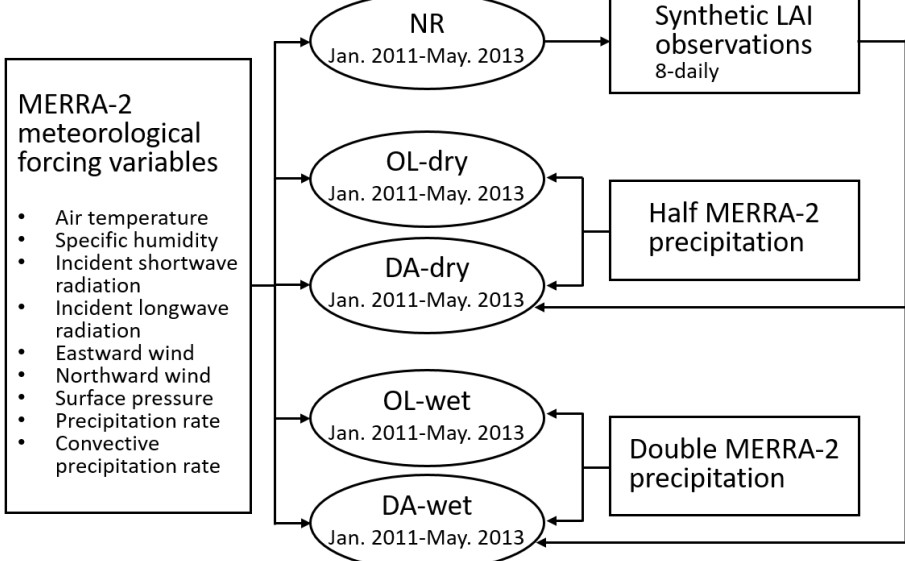


Figure 2. Schematic diagram of the OSSE design.

Figure 2 shows a schematic diagram of the experiments. First, the Noah-MP model is spun-

up for a 10-year period (2001-2010) to ensure a physically realistic state of equilibrium. Second,
the model is run for a 29-month period (January 2011 – May 2013) to conduct the Nature Run
(NR) with the same configuration as the spin-up one. Output from the NR represents the "truth"
in the OSSE. The LAI outputs from NR serve as synthetic LAI observations (after being perturbed



via an error model) for the DA runs. Third, two Open Loop (OL) runs (no DA) are conducted for
the same 29-month period under two conditions: i) "extremely dry" condition (the model is forced
by halving the MERRA-2 precipitation data; OL-dry), and ii) "extremely wet" condition (the
model is forced by doubling the MERRA-2 precipitation; OL-wet).

The two DA runs are then produced under the same conditions (DA-dry and DA-wet) using

an EnKF assimilation algorithm. The synthetic LAI observations are assimilated to the system at
8-daily frequency. The synthetic LAI observation has the same temporal resolution as the MODIS
LAI product but with full coverage over the entire study domain. In real case studies, satellite LAI
products contain a substantial amount of missing data mainly due to the cloud obscuration gaps.
Based on the vegetation type in the model, the leaf mass fields are also updated. Random
perturbations of MERRA-2 meteorological forcings and synthetic LAI observations are applied to
create an ensemble of land surface conditions that represent the uncertainties of LSM.

Similar to previous work (Kumar et al. 2014, 2018, 2019), the MERRA-2 shortwave and

longwave radiations as well as precipitation are perturbed hourly. Multiplicative perturbations are
applied to the shortwave radiation and precipitation with a mean of 1 and standard deviations of
0.3 and 0.5, respectively. The longwave radiation is perturbed via an additive perturbation with a
standard deviation of 50 W/m$^2$. The perturbations of the three meteorological forcing variables
also include cross correlations: cross correlation between shortwave radiation and precipitation is
-0.8, cross correlation between longwave radiation and precipitation is 0.5; and cross correlation
between shortwave and longwave radiations is -0.5. The synthetic LAI observations are perturbed
via an additive model with a standard deviation of 0.1.

To select the optimal ensemble size, a sensitivity test is performed for ensemble sizes

spanning from 2 to 24 members (not shown here). The number of ensemble members has a strong




impact on the model results at small sizes, while the model performance tends to become steady
when more than 20 ensemble members are considered. Thus, all the DA simulations were run for
20 members.

*2.3. Evaluation and error metrics*
Output variables from the OL and DA runs are evaluated against the "truth" from the NR at daily,
monthly, and seasonal temporal scales. Besides LAI, five more water fluxes and storages are
evaluated in the results section: evapotranspiration, interception evaporation, canopy water storage,
surface soil moisture, and terrestrial water storage.

The first 5-month model outputs are eliminated from the evaluation to avoid model

systematic instability at the beginning of the DA simulations and the evaluation, thus, focused only
on model outputs from 2011-06-01 to 2013-05-31. Results are discussed using both maps and
anomaly time series. Each of the anomaly time series is computed relative to its respective model
run. Moreover, two error metrics are employed to quantify the difference between OL (and DA)
with respect to the reference variables (from the NR). The first one is the Normalized and Centered
Root Mean Square Error (NCRMSE), defined as follows:
$$E = \frac{\left\{\frac{1}{N}\sum_{i=1}^{N}[(X_i - mean(X)) - (O_i - mean(O))]^2\right\}^{\frac{1}{2}}}{mean(O)}$$   Eq. 1
where $E$ is the NCRMSE, $O$ is the NR output variable, and $X$ is the output variable from the OL
runs or DA runs. Second, to investigate the improvement (or degradation) due to the DA of LAI
observations, we adopt the Normalized Information Contribution (NIC) index based on NCRMSE
and defined as:
$$C = \frac{E_{DA} - E_{OL}}{0 - E_{OL}}$$   Eq. 2





where *C* represents the NIC index and *E* is the NCRMSE for OL or DA runs.. NIC equals to 1
means that DA realizes the maximum possible improvement over the OL; NIC equals to zero
means that DA and OL show the same performance skills; and negative NIC indicates a model
degradation through DA.

**3. Results and discussion**
*3.1. LAI*

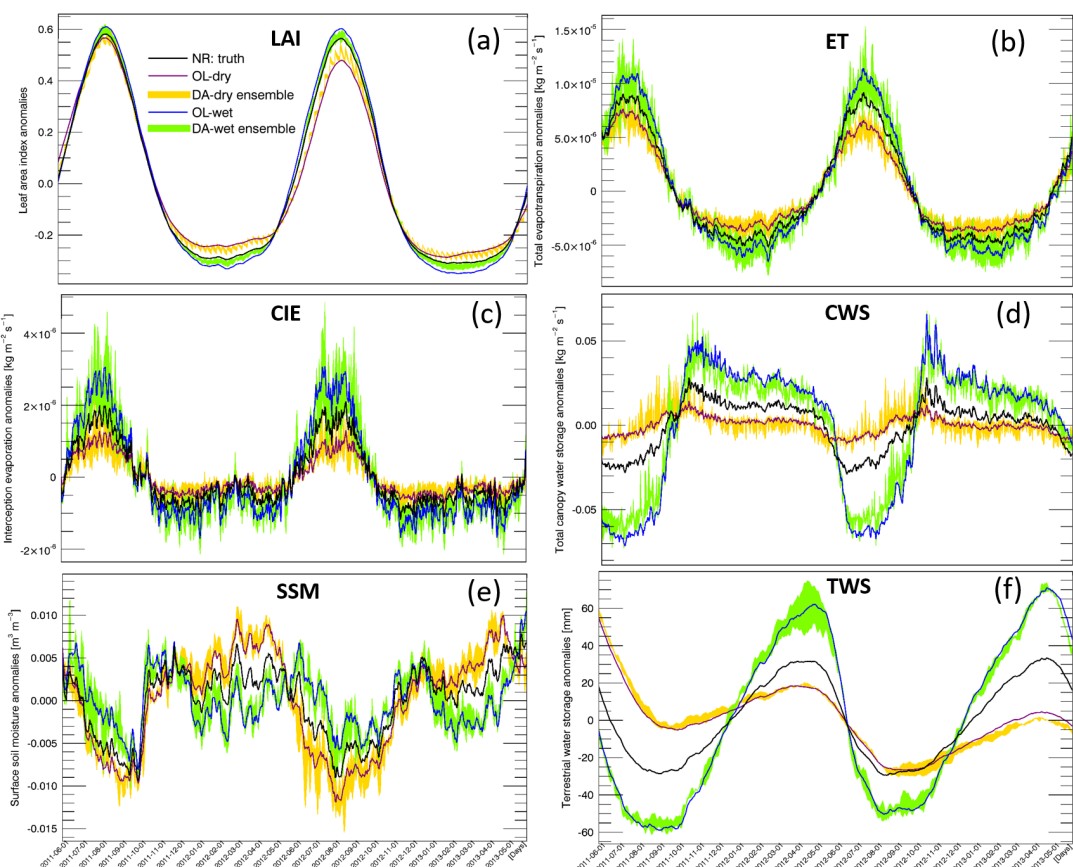


Figure 3. Global averaged daily anomalies of LAI and five water variables (2011-06-01 to 2013-05-30).






Figure 3 (a) shows time series of global averaged LAI anomalies. As expected, LAI anomalies are
largely impacted by the extreme precipitation conditions. Moreover, the seasonality of LAI
anomalies is evident, showing larger variations in winter and summer than during the transition
periods (spring and fall). The OL-wet condition simulation (blue line) shows larger LAI anomalies
than the NR reference (black line), while the OL-dry condition (purple line) has smaller LAI
anomalies than NR. The green and yellow shaded areas represent the 20 ensemble members of the
DA runs. The LAI DA procedure under both wet and dry conditions effectively corrects the LAI
anomalies comparing to the reference anomalies. In general, DA performs better in the wet
condition experiment than in the DA-dry case.
Moreover, DA runs show lower NCRMSEs than the corresponding OL runs in several
regions across the globe (Figure 4a), with larger over shrubland and grassland areas (refer to Figure
1 for land covers).
In order to illustrate how LAI assimilation performs for different seasons, Figure 5a and
Figure 6a show monthly averages of NCRMSE for LAI across the northern and southern
hemispheres, respectively. In the northern hemisphere (Figure 5a), the NCRMSE time series
follow clear seasonal patterns. First, the NCRMSE is higher in winter/spring and is lower in
summer/fall for both extreme precipitation conditions. The highest NCRMSE values are in March
and April (spring), and the lowest values are in July, August, and September. The differences of
NCRMSE between OL and the corresponding DA runs tend to be much larger in spring than in
any other seasons, which means that LAI assimilation is more effective in the vegetation growth
period. Moreover, the NCRMSE is constantly higher in the dry condition runs than the wet ones,
which is due to the fact that the growth of vegetation is sensitive to the lack of water. Differences
between wet and dry conditions are much smaller in summer than in other seasons. In the summer,





the vegetation leaves in the north hemisphere are fully developed and the plants can use stomatal
closure to preserve water under water limited condition (dry condition), thus the NCRMSE of dry
condition becomes smaller and does not show much difference from the wet condition. The
southern hemisphere (Figure 6a), which does not have a strong climate seasonality, shows more
modest seasonal NCRMSE patterns than the northern regions. In general, the NCRMSEs in the
southern hemisphere are smaller than the ones in the northern hemisphere all year around.
Specifically, NCRMSEs in the southern hemisphere are slightly higher in October, November, and
December, when the differences between OL and DA runs are also larger.

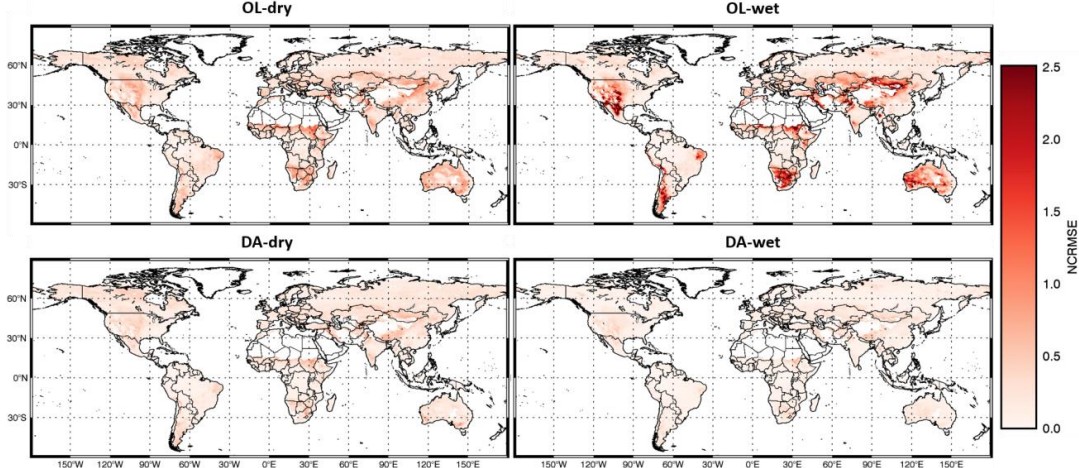

Figure 4. Maps of LAI NCRMSE for the OL and DA runs.


### 253   *3.2. Water fluxes and storages*

As mentioned in section 2.3, we focus on five water-related variables from the Noah-MP output
to evaluate the impact of LAI assimilation on simulating the water cycle (ET, CIE, CWS, SSM,
and TWS). Daily time series of global anomalies of the five water variables are shown in Figure
3(b-f). The model well simulates the seasonality of anomalies for all water fluxes/storages
considered here. The OL runs reveal that all the five variables are impacted by the highly biased



precipitation conditions (dry and wet). Specifically, the variations of ET, CIE, CWS, and TWS
tend to be amplified by the wet condition and tend to be dampened by the dry condition. On the
contrary, the anomalies of SSM become larger in dry conditions and become smaller in wet
conditions, which is probably due to the limited soil water capacity. The surface soil has higher
chance to get saturated in wet conditions when the precipitation doubles the original amount, but
SSM cannot get larger once the soil is saturated, even if there is more precipitation added to the
system. Thus, the range of SSM anomalies in the wet experiment is limited and narrower than in
the dry condition. The green and yellow shaded areas represent the ensemble of the DA runs. The
anomaly ensembles of the five water variables show slight improvements through DA when
precipitation is severely positively biased (wet condition). However, none of these variables shows
improvement when the precipitation is severely negatively biased (dry condition) – the anomalies
either have no change through the LAI DA (ET, CIE, and CWS) or worsen the OL-dry run (SSM
and TWS).

To further investigate the efficiency of assimilating LAI in Noah-MP, time series of

monthly NCRMSE averages are shown in Figure 5(b-f) and Figure 6 (b-f) for all five water
variables. The five variables can be divided into two main groups based on their performances:
ET/CIE/CWS and SSM/TWS. For the wet bias experiment, DA improves the NCRMSE for all
variables. However, LAI assimilation is not able to correct the model when the input precipitation
is negatively biased (dry condition) and the NCRMSEs of DA runs are either the same as in the
OL runs (ET/CIE/CWS) or worse (SSM/TWS). Specifically, ET/CIE/CWS have larger NCRMSE
in the northern hemisphere and much smaller NCRMSEs in the southern hemisphere, but
SSM/TWS do not show large differences between north and south. Moreover, ET/CIE/CWS in
the northern hemisphere follow a seasonal pattern: NCRMSEs are lower in summer and higher in





the colder seasons (December, January, February, and March). In the southern hemisphere the
three variables also have relative higher NCRMSE in the colder season (June, July, and August).
On the contrary, SSM/TWS show a different seasonal pattern that NCRMSEs are larger in the
warmer season (April, May, and June) over northern hemisphere. In southern hemisphere, TWS
also has larger NCRMSEs in warmer season (October to April), but SSM shows higher NCRMSEs
in colder season (similar to the ET/CIE/CWS group).

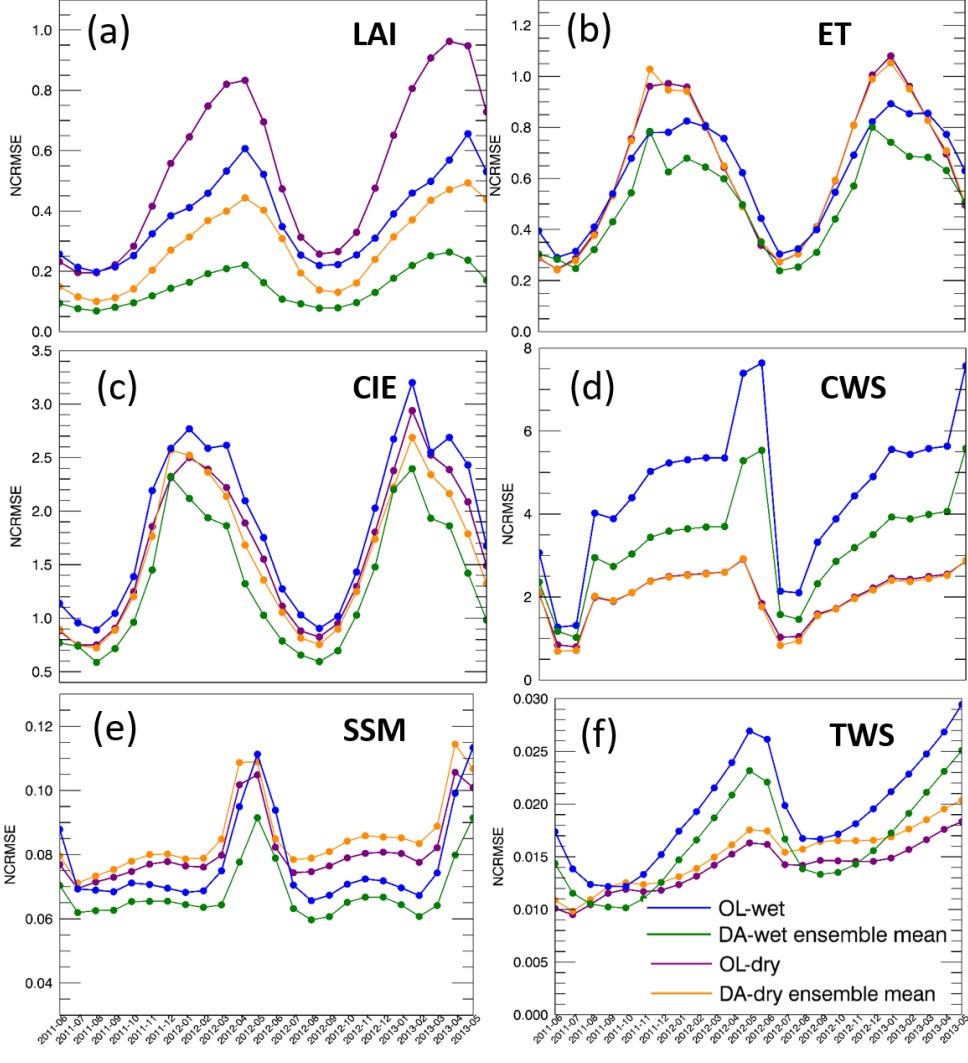


Figure 5. Monthly averaged NCRMSE for LAI and five water variables over the Northern hemisphere.

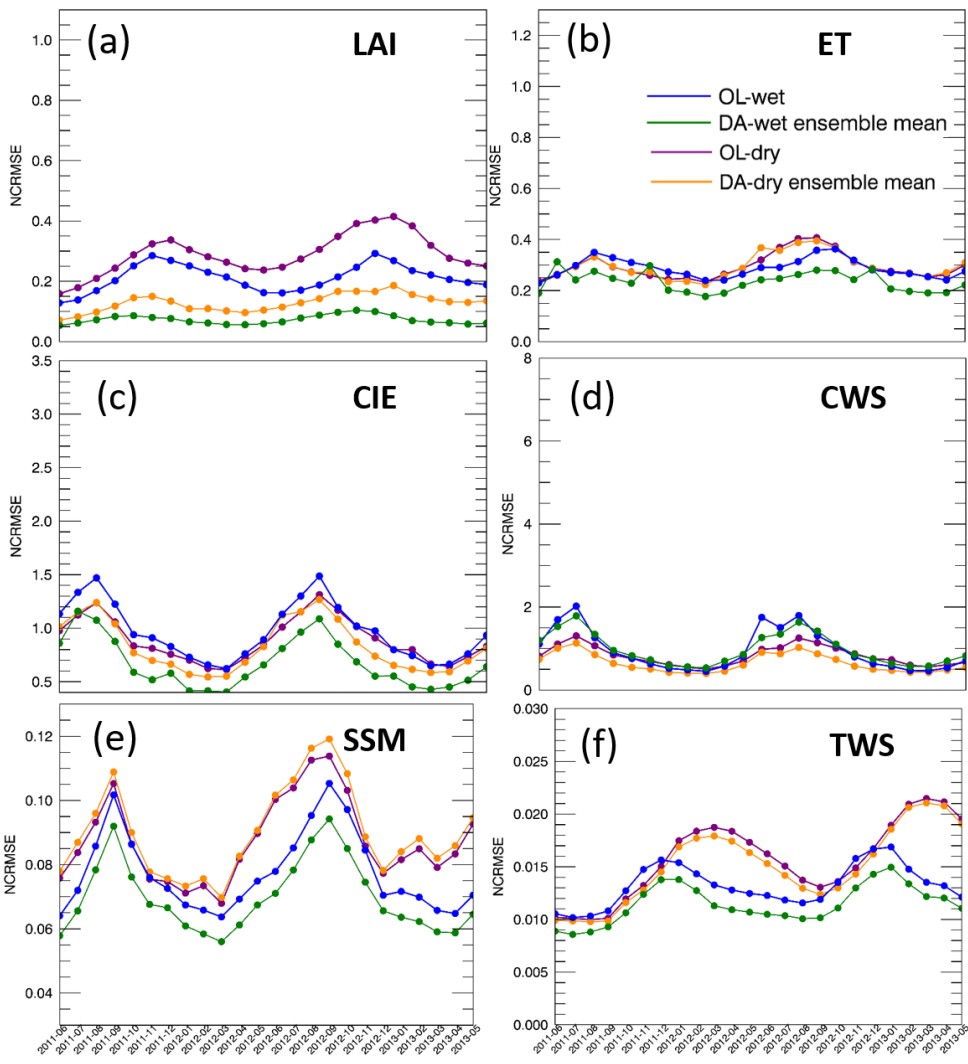

Figure 6. Same as in Figure 5, but for the Southern hemisphere.

The improvements in the model water fluxes and storages through LAI DA are also
quantified by the NIC index (defined in Eq. 2). Figure 7 presents comparisons among NIC indices
for each water variable analyzed in this study across areas with four different land cover types:
forest & woodland, grassland, shrubland, and cropland. In general, LAI DA improves the NIC
indices with positively biased input precipitation (wet condition) but worsens the NIC when
negatively biased input precipitation (dry condition) is considered. Specifically, in wet condition,





ET, CIE, and CWS have higher variability over areas with different land cover types, while SSM
and TWS have similar NIC values across different land covers. Shrubland and cropland tend to
perform better in wet condition except for TWS. In dry condition, the NICs of ET, CIE, and TWS
have higher variability than the ones of CWS and SSM. SSM and TWS show very low NIC values
in dry condition for almost all land covers. Overall the NIC values of ET, CIE, and CWS are better
than the ones of SSM and TWS for all land cover types, though the NICs of ET and CIE over
forest & woodland perform very poorly.

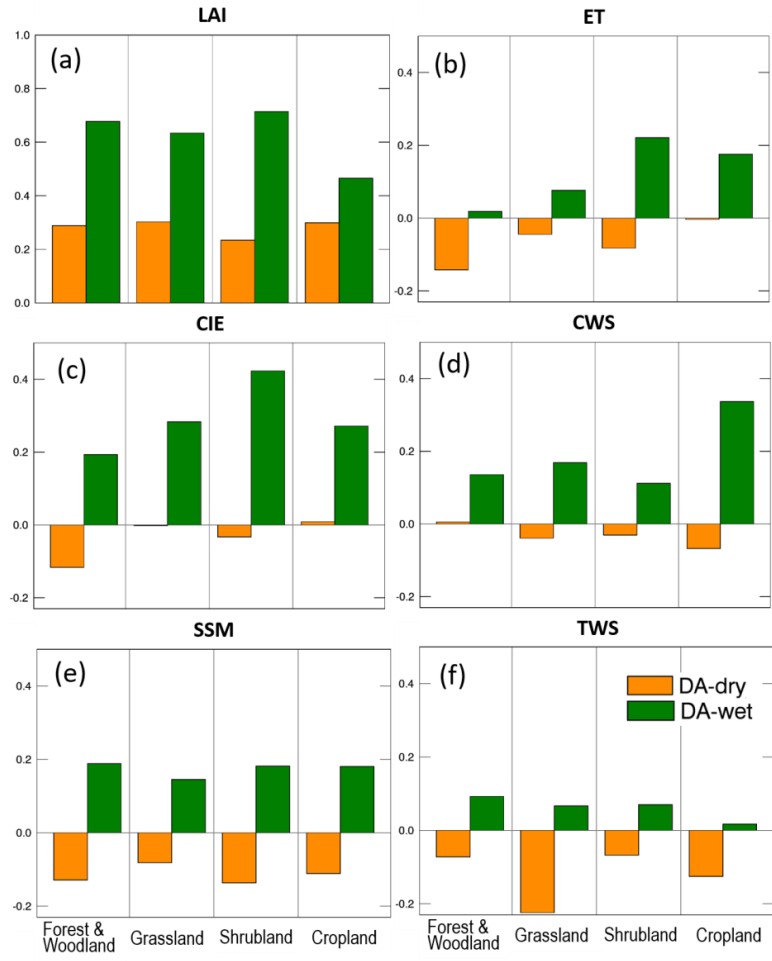


Figure 7. NIC for different variables and different land cover types for the two DA runs.



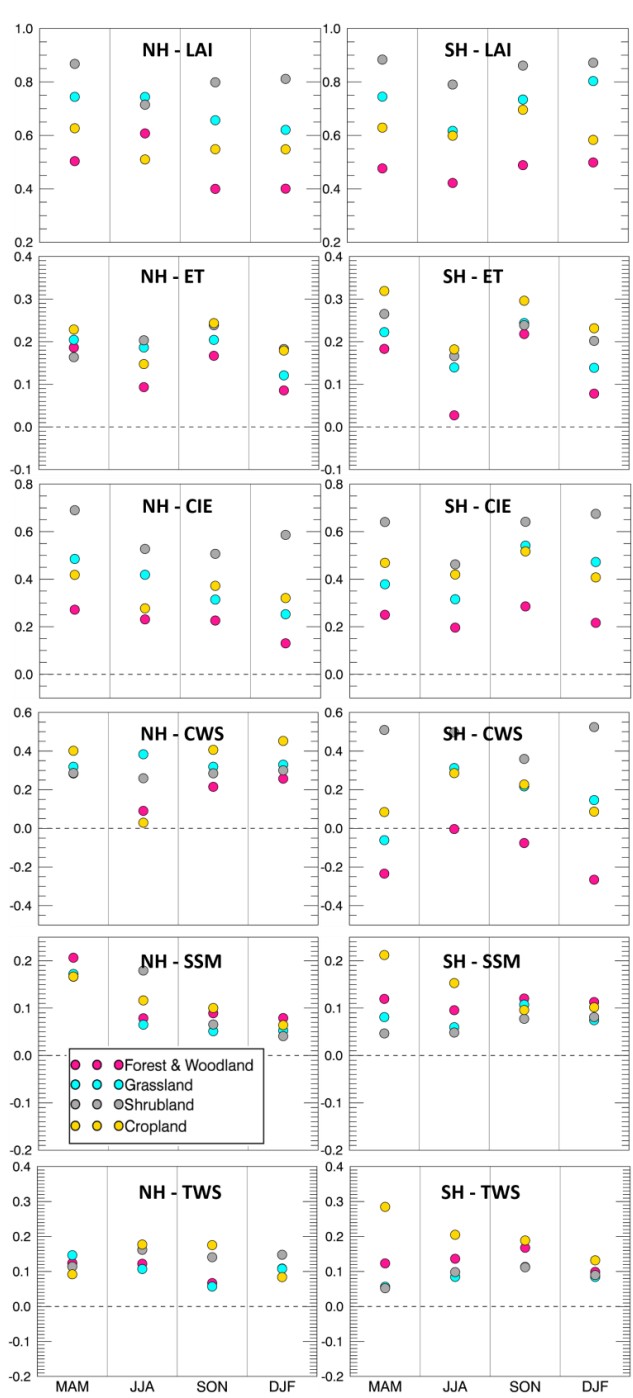


Figure 8. NIC of five water variables under wet precipitation conditions over northern and southern hemispheres


(NH and SH) during different seasons (MAM, JJA, SON, and DJF)

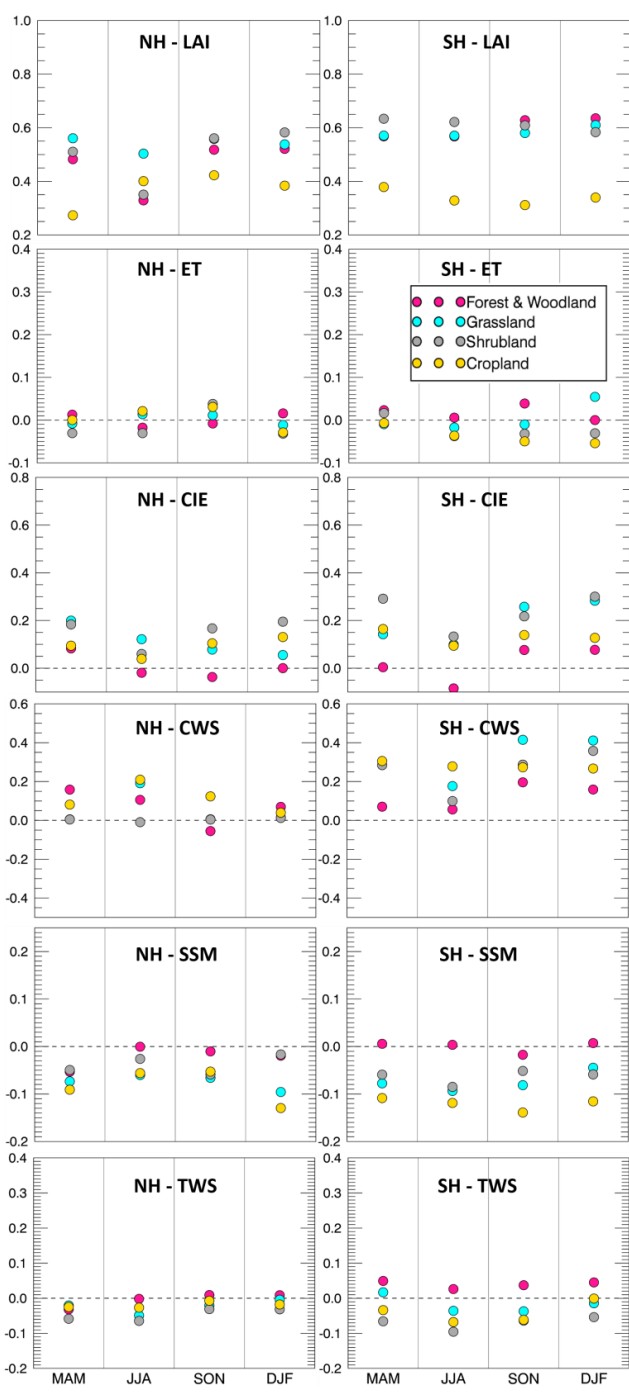


Figure 9. Same as in Figure 8, but for the dry precipitation experiment.





The effectiveness of LAI DA therefore varies across the northern and southern hemispheres,
different land cover types, as well as different input precipitation biases. To further investigate the
influence of LAI assimilation, Figures 8 and 9 present NIC values for each hemisphere, each
season, and each of the input precipitation conditions – wet and dry, respectively. For the wet case
(Figure 8), NIC is positive in most cases, which means that the five water variables benefit from
the LAI assimilation in all seasons and in both hemispheres. The only exception is CWS which
has negative NIC values in the southern hemisphere over grassland (in MAM season) and over
forest & woodland (in all seasons). In fact, the forest & woodland region tends to have the least
improvement through the LAI assimilation among all land cover types. This is probably because
forests and woodlands have large water-holding capacity; thus, the change of water amount caused
by LAI DA is not enough to improve the water-related variables. In other words, forest and
woodland areas tend to have lower sensitivity in response to the change of precipitation conditions.
On the contrary, cropland is very sensitive to precipitation and it benefits the most from the
assimilation of LAI for most of the variables. Moreover, NICs of ET/CIE/CWS tend to be smaller
than the NICs of SSM and TWS. There is no clear seasonality in the NIC values, though it has a
weak tendency to be lower in warm seasons.
For the dry condition case (Figure 9), NIC values are much lower than in the wet bias case.
Nearly half of the NIC values for the five water-related variables are negative, meaning that DA
degrades the OL estimates. Nevertheless, the forest & woodland regions tend to perform better
than other land covers in dry condition for SSM and TWS. This is due to large water-holder
capacity of forests and woodlands, which keeps the model water storage more stable when the
input precipitation is affected by large negative biases.





### *3.3. Discussion*

Results presented in sections 3.1 and 3.2 indicate that assimilating LAI in Noah-MP improves the model estimates of water fluxes and storages under positively biased precipitation input (wet case), but does not benefit most of the selected water variables when the precipitation input is characterized by a negative bias (dry case).

In the dry condition runs, Noah-MP is fed by only half of the original MERRA-2 precipitation used in the NR. Considering that the amount of water in Noah-MP is conservative (since based on a water balance equation), the model has no additional water source in the system, even though the LAI assimilation pushes the model towards more vegetation (that should result in more water). As a matter of fact, introducing more vegetation in the system results in more evapotranspiration and more root water uptake from the soil, which is most likely the cause for the poor performance of most water fluxes and storages in the DA-dry experiment.

On the other hand, the LAI assimilation is found to improve the original OL runs when the input precipitation is positively biased (DA-wet vs. OL-wet). This is because LAI assimilation is able to help constrain the partitioning of model water storage when there is abundant water in the system, thus, improving the performance of water-related variables. In summary, although the EnKF is run here in a sub-optimal mode (not satisfying the unbiasedness assumption), the assimilation of LAI is shown to have a positive impact on multiple variables and in several regions of the world.

## 4. Conclusions

This study evaluates the efficiency of assimilating vegetation information (i.e., LAI synthetic observations) within a land surface model when the precipitation forcing data are strongly biased





(either positively or negatively). Two OSSEs that use an EnKF algorithm for LAI assimilation are
performed at the global scale during June 2011 – May 2013. The experiments use Noah-MP as a
land surface model and MERRA-2 as meteorological forcing data. The OL and DA runs are
evaluated against a synthetic "truth" from a nature run, in which the MERRA-2 precipitation is
neither perturbed nor biased. The performance of the proposed framework is evaluated for several
model output, including LAI estimates and five water-related variables (ET, CIE, CWS, SSM, and
TWS).

Overall the EnKF LAI assimilation procedure effectively reduces the LAI error under

positively (wet case) and the negatively (dry case) biased precipitation conditions. For the five
selected water flux or storage variables, LAI DA improves the model estimates when the model
input precipitation is positively biased (wet), but tends to worsen the OL estimates for some of
those variables when the input precipitation is negatively biased (dry). Specifically, SSM and TWS
estimates are degraded in the DA-dry run with respect to the OL-dry run, while ET, CIE, and CWS
do not present large changes when LAI is assimilated in the dry bias run. The poor performance
of LAI DA under dry condition is mainly attributed to the fact that the amount of water in Noah-
MP is conservative. The LAI assimilation in dry condition introduces more vegetation, which
requires more water in the system to replenish the soil water supply. However, the model has no
additional source of water, since the input precipitation is negatively biased.

Although a blind bias case (e.g., unknown biases in the precipitation forcing dataset) is

presented here in which the EnKF is run in a sub-optimal mode, the assimilation of LAI
observations is proven useful to improve several model output variables. Future research should
focus on alternative methods to run the DA system in a more optimal way such as updating other
related model states while assimilating LAI observations and on the assimilation of actual satellite-





based LAI observations (e.g., MODIS, GLASS) at the global scale to verify the efficiency of
vegetation DA on water cycle variables. This may be particularly useful in agricultural areas,
where the vegetation conditions are largely impacted by cropping schedules (Kumar et al. 2019).
Moreover, future work should investigate multi-variate DA techniques that combine the
assimilation of several variables (such as LAI, soil moisture, and TWS) at once.

*Acknowledgements:* This research is sponsored by the NASA Modeling, Analysis, and
Prediction (MAP) Program (80NSSC17K0109). We would also like to acknowledge the
computational resources and support from the ARGO HPC Cluster team at George Mason
University.





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
