# Peer review of "1. Introduction"

_Hydrology and Earth System Sciences, 2019_

## Referee Comment (RC1) · Anonymous Referee #1 · 11 Nov 2019

article amsmath setspace

**General comments**

The authors aim to assess to what extent the Noah-MP model can be optimized through the assimilation of leaf area index (LAI) observations at global scale. By utilizing two observing system simulation experiments (OSSEs) and the EnKF algorithm, the efficiency of assimilating LAI and model performance for water related variables are discussed.

At first in my opinion this manuscript needs to be proofread/revised carefully for academic writing.

[Figure]

Something that I do not understand is that the authors use the simulated LAI from the nature run as the 'truth' instead of observations. If nature run can achieve the "truth", why did the authors conduct assimilation based on different conditions (wet or dry)?

Other important comment is that why did the authors use the precipitation which are extremely biased instead of using a more precise precipitation forcing. Furthermore, did the authors run the assimilation experiment using the MERRA-2 precipitation instead of halving or doubling the value?

In conclusion, the manuscript in its current form suffers from several issues that prevent it to be published as is. In my opinion the paper still worth to be published after addressing all these issues, and a major revision is asked.

**Specific comments**

1. P3L56-57: As far as I know, LSMs not only couple with dynamic vegetation models, but also involve some dynamic vegetation modules. So the statement is not appropriate.

2. Section 2.2: Why do you use the precipitation forcing data which are strongly biased.

3. Why did you choose the LAI simulations from the nature run as the "truth" instead of using the LAI observations? As you have described the reasons from P9L171 to L172, there are many other LAI products without missing data which can be used for assimilation.

4. Did you evaluate the LAI or other variables from the natural run by using remote sensing LAI datasets or other kinds of observations?

5. P9L178-P9L184: How did you determine the values of multiplicative perturbations (such as, the shortwave radiation and precipitation with a mean of 1 and standard deviations of 0.3 and 0.5, the standard deviation for longwave radiation of 50 W/m2, the standard deviation for LAI of 0.1)?

6. Have the evaluation and error metrics been used in former studies? If so, please list at least one references.

7. How did you determine the initial conditions?

8. The discussion section should include the discussion of the results in the context of other papers dealing with the same of similar subjects.

9. A more in-depth analysis of the results is necessary. In this paper the authors only talk about the statistical characteristic variables (such as the NCRMSE, NIC, etc) of LAI and water related variables. Why not focus on the LAI and water related variables themselves?

10. Why only perturb the meteorological forcing and not the initial conditions and/or model parameters?

11. How sensitive is LAI with respect to the meteorological forcing?

**Technical corrections**

1. P2L27-L28: Can you illustrate which land surface model you use here? And the same to P2L38, P5L104, and so on.

2. P2L28-L29: Remove "the" from the phrase of "at the global scale", and the same to P5L100, P5L100, P22L361, and so on.

3. P3L44: Do not need to leave two blank spaces here.

4. P3L46: It's not appropriate to use "between" among vegetation, precipitation, and soil moisture.

5. P3L51: The related references cited here are not enough to illustrate the phenomenon that "these land surface processes and feedbacks have been examined through numerical modeling experiments". List more. . .

6. P3L54: You needn't capitalizes the first letter for leaf area index.

7. P4L67: "the Moderate Resolution Imaging Spectroradiometer" has been abbreviated to "MODIS" before.

8. P4L88-P5L90: Please refine this sentence.

9. P5L95: Change "model simulated LAI" to "simulated LAI".

10. P5L97: Please refine the statement of "focused on small regions".

11. P5L106-L107: Please define the abbreviation of all the water related variables when they first appear in this manuscript. Furthermore, "evapotranspiration" has been abbreviated to "ET" in P5L93.

12. P5L110: Please specify which land surface model.

13. P6L116-120: Please refine this sentence as it is too long.

14. P6L121: Please define "NASA".

15. P6L126: Keep the tense consistent.

16. P6L133-P7L138: Please define the abbreviation of all the water related variables when they first appear in this manuscript.

17. P7L150: I am not sure whether the state of "a LAI EnKF" is appropriate.

18. P7L153: The phase of "on a global scale" is not appropriate.

19. P10L188-L189: Keep the tense consistent.

20. P10L194-L195: The water related variables have been defined before, and you can use their acronyms.

21. P10L203: What does i and N in Equation 1 mean?

22. P10L208: the word "O" in the denominator looks like "zero" in Equation 2.

23. P11L209: There are two periods.

24. P12L220-L222: As Figure 3 shows the GLOBAL averaged LAI anomalies, it is better to use the statement of month (or JJA and SON seasons) instead of winter/summer season.

25. P12L229: Please refine this sentence.

26. P12L241: Remove "the". Furthermore, this sentence is a little too long in my opinion.

27. P14L263: Please change the "has higher chance" into "is more likely to".

28. P14L268-269: I think this is the first appearance that positively biased is wet condition (or negatively biased is dry condition), or maybe earlier, and this statement does not need to be repeated each time it appears in this paper (see P14L277, P16L296-L297, P21L337, P21L339).

29. P15L282-L287: It is better to use the statement of month instead of season.

30. Please add the description for the Y-coordinate for Figure 7, 8 and 9.

31. P21L357: Please specify which land surface model.

---

## Referee Comment (RC2) · Anonymous Referee #2 · 5 Dec 2019

General comments:

Synthetic observations are used to assess the impact of assimilating satellite-derived LAI estimates into the Noah land surface model. A major shortcoming of the assimilation system used in this study is that LAI assimilation has no direct impact on soil moisture. As a result, dry precipitation biases cannot be compensated for. This issue was at least partly solved in other assimilation systems. Unfortunately, the relevant literature is not completely cited. This paper is not well written, not complete for understanding, and cannot be published in the present form. Methods description is incomplete. Interpretation of results is made in the Result section instead of the Discussion

section.

Recommendation: major revision.

Particular comments:

- L. 39-4: Examples of joint assimilation of LAI and soil moisture in a land surface model can be found in the literature.

- L. 95-97: In the same context and at the continental scale, Albergel et al. showed that sequential LAI assimilation can be used to analyse soil moisture at various depth, in addition to vegetation biomass (https://doi.org/10.5194/gmd-10-3889-2017). This property is particularly useful in dry conditions, when surface soil moisture tends to be decoupled from deeper soil layers.

- L. 109 (Section 2): A section describing the DA method is needed. What are the analysed variables? Does LAI DA impacts soil moisture?

- L. 138: How are subsurface waters represented? Do you represent inundations plains? Lakes?

- L. 188 (ensemble members): How is this ensemble generated?

- L. 196-197: Why are these instabilities generated by DA?

- L. 203 (Eq. 1): Why do you use NCRMSE and not standard score metrics such as RMSE or ubRMSE (i.e. standard deviation of differences)?

- L. 217 (Figure 3): Please change evaporation units. Since these time series are daily, should be per day instead of per second. It seems that CWS anomalies are 3 order of magnitude larger than ET anomalies. Why? Define here what you mean by "anomaly" (not defined in the text). NR anomalies: with respect to what? Is NR the benchmark or not? Real values have to be showed at some stage. Not only anomalies.

- L. 243 ("thus the NCRMSA . . . becomes smaller"): Why?

- L. 276-277 (LAI assimilation unable to correct for dry precipitation bias): Why?

- L. 320-322: I don't see the logics. I would expect that large water-holding capacity would enhance the impact of LAI DA.

- L. 323 (forests and woodlands): Is this because of large rooting depth?

- L. 331: Water-holding? Do you mean interception reservoir or soil reservoir?

- L. 374-375: This could be because the used DA system is not able to analysed RZSM from LAI observations. Please explain.

Editorial comments:

- L. 251 (Figure 4): Color scale is difficult to interpret. Please use several colors (e.g. blue in addition to red).

- L. 289 (Figure 5): Time axis labels are not readable. Please improve!

- L. 291 (Figure 6): Time axis labels are not readable. Please improve!

---

## Author Comment (AC2) · 20 Feb 2020

The authors would like to thank the reviewer for their time, effort, and detailed comments. All suggestions have been incorporated into the manuscript and explained in this document. We also thoroughly proofread and revised the whole manuscript.

**General comments**
Synthetic observations are used to assess the impact of assimilating satellite-derived LAI estimates into the Noah land surface model. A major shortcoming of the assimilation system used in this study is that LAI assimilation has no direct impact on soil moisture. As a result, dry precipitation biases cannot be compensated for. This issue was at least partly solved in other assimilation systems. Unfortunately, the relevant literature is not completely cited.

This paper is not well written, not complete for understanding, and cannot be published in the present form. Methods description is incomplete. Interpretation of results is made in the Result section instead of the Discussion section.

**Recommendation: major revision.**

**Particular comments:**
- L. 39-40: Examples of joint assimilation of LAI and soil moisture in a land surface model can be found in the literature.

We have reviewed several studies that used LAI-SM joint assimilation (Pauwels et al. 2007; Sabater et al. 2007; Barbu et al. 2011; Fairbairn et al. 2017) and cited them in the discussion section:

*"Overall the improvement of water variables through LAI assimilation is not remarkable enough to compensate the model degradation caused by the biased precipitation forcing data. Previous studies (Pauwels et al. 2007; Sabater et al. 2007; Barbu et al. 2011; Fairbairn et al. 2017) have tested the performance of the joint assimilation of LAI and soil moisture over regional domains and showed promising results. However, no experiment was performed at the global scale. Future work could investigate a multi-variate data assimilation system that concurrently merges both LAI and soil moisture (or TWS) observations globally."*

All the cited LAI-SM joint DA studies were conducted over regional domains. We emphasized "at global scale" in the last sentence of the abstract to make the statement more accurate:

*"Future work could investigate a multi-variate data assimilation system that concurrently merges both LAI and soil moisture (or TWS) observations at global scale."*

- Barbu, A. L., Calvet, J. C., Mahfouf, J. F., Albergel, C., & Lafont, S. (2011). Assimilation of Soil Wetness Index and Leaf Area Index into the ISBA-A-gs land surface model: grassland case study. Biogeosciences, 8(7), 1971-1986.
- Fairbairn, D., Barbu, A., Napoly, A., Albergel, C., Mahfouf, J. F., & Calvet, J. C. (2017). The effect of satellite-derived surface soil moisture and leaf area index land data assimilation on streamflow simulations over France. Hydrology and Earth System Sciences, 21(4), 2015-2033.
- Pauwels, V. R., Verhoest, N. E., De Lannoy, G. J., Guissard, V., Lucau, C., & Defourny, P. (2007). Optimization of a coupled hydrology–crop growth model through the assimilation of observed soil moisture and leaf area index values using an ensemble Kalman filter. Water Resources Research, 43(4).
- Sabater, J. M., Rüdiger, C., Calvet, J. C., Fritz, N., Jarlan, L., & Kerr, Y. (2008). Joint assimilation of surface soil moisture and LAI observations into a land surface model. Agricultural and forest meteorology, 148(8-9), 1362-1373.

- L. 95-97: In the same context and at the continental scale, Albergel et al. showed that sequential LAI assimilation can be used to analyse soil moisture at various depth, in addition to vegetation biomass (https://doi.org/10.5194/gmd-10-3889-2017). This property is particularly useful in dry conditions, when surface soil moisture tends to be decoupled from deeper soil layers.

Thank you for providing this reference. We added it to the manuscript. *"Only a few studies discussed the influences of LAI assimilation on the estimation of water variables such as soil moisture or streamflow (Pauwels et al. 2007; Sabater et al. 2008) and most of them focus on limited regions. Most recently, Albergel et al. (2017) conducted a study on a much larger domain–Europe and the Mediterranean basin–and showed that LAI assimilation can be used to improve soil moisture at various depths."*

- Albergel, C., Munier, S., Leroux, D.J., Dewaele, H., Fairbairn, D., Barbu, A.L., Gelati, E., Dorigo, W., Faroux, S., Meurey, C. and Le Moigne, P.: Sequential assimilation of satellite-derived vegetation and soil moisture products using SURFEX_v8. 0: LDAS-Monde assessment over the Euro-Mediterranean area, Geosci. Model Dev., 10, 3889-3912, https://doi.org/10.5194/gmd-10-3889-2017, 2017.

- L. 109 (Section 2): A section describing the DA method is needed. What are the analysed variables? Does LAI DA impacts soil moisture?

The DA method in this study is implemented in the NASA Land Information System (LIS). The method has been applied in many sequential data assimilation studies. We added references for the DA method in section 2.2:

*"The two DA runs are then conducted under the same conditions (DA-dry and DA-wet) using a one-dimensional EnKF assimilation algorithm which is a built-in DA method in LIS. The EnKF DA algorithm is suitable for non-linear and intermittent land surface processes (Reichle et al. 2002a, b). Details of the algorithm have been illustrated in previous studies (Reichle et al. 2010; De Lannoy et al. 2012; Liu et al. 2015; Kumar et al. 2019)."*

- L. 138: How are subsurface waters represented? Do you represent inundations plains? Lakes?
TWS is the sum of snow water equivalent, surface water, soil moisture, and groundwater. So, subsurface water (i.e., groundwater) is included. Lakes and inundation plains are considered as surface water, which is also included in TWS.

- L. 188 (ensemble members): How is this ensemble generated?
The model ensemble is generated by perturbing the meteorological forcing inputs (precipitation and shortwave/longwave radiations). The figure below shows the ensemble size sensitivity test. The model performance tends to become steady when more than 20 members are considered, which is why all the DA simulations run for 20 members.

[Figure]

A detailed description can be found in section 2.2 of the manuscript.
"*The model ensemble is generated by perturbing a set of meteorological forcing. To select the optimal ensemble size, a sensitivity test is performed for ensemble sizes spanning from 2 to 24 members (not shown here). The number of ensemble members has a strong impact on the model results at small sizes, while the model performance tends to become steady when more than 20 ensemble members are considered. Thus, all the DA simulations are run for 20 members.*"

- L. 196-197: Why are these instabilities generated by DA?
The initial condition of OL and DA runs is generated by a 10-year spin-up run which uses the original MERRA-2 precipitation as metrological forcing. The OL and DA runs are forced by either doubled or halved precipitation that is not consistent with the spin-up run. So, the model needs to run for a certain time before stabilizing. The figure below shows the global averaged LAI time series from the beginning of the simulation (Jan. 1st, 2011) to Dec. 31st, 2011. The LAI simulated by OL and DA runs does not get stable until around May. Therefore, we decided to eliminate the first 5-month model outputs in the analyses.
    We added this explanation in the manuscript in section 2.3. "*The initial condition of OL and DA runs is generated by a spin-up run that uses the original MERRA-2 precipitation as input. However, the OL and DA runs are forced by either doubled or halved precipitation, which is not consistent with the spin-up run and the model needs some time to stabilize. The first 5-month model outputs are therefore eliminated from the evaluation to avoid the model systematic*

*instability at the beginning of the OL and DA simulations and the evaluation, thus, focused only on model outputs from 2011-06-01 to 2013-05-31."*

[Figure]

- L. 203 (Eq. 1): Why do you use NCRMSE and not standard score metrics such as RMSE or ubRMSE (i.e. standard deviation of differences)?
In the manuscript, we compared the result of LAI and five water related variables (ET, CIE, CWS, SSM, and TWS). Units of these variables are very different, which is why we decided to adopt unitless statistical metrics. UbRMSE is certainly another valid option.

- L. 217 (Figure 3):
(1) Please change evaporation units. Since these time series are daily, should be per day instead of per second.
(2) It seems that CWS anomalies are 3 order of magnitude larger than ET anomalies. Why?
(3) Define here what you mean by "anomaly" (not defined in the text).
(4) NR anomalies: with respect to what?
(5) Is NR the benchmark or not?
(6) Real values have to be showed at some stage. Not only anomalies.

1) The ET value showed in Figure 3 is the model output, which is an average over the day. So, the unit is "kg m$^{-2}$ s$^{-1}$"
2) CWS is the canopy water storage which include the water stored in the leaves and the intercepted water. So, it is much larger than the ET.
3) Anomalies are defined in section 2.3. "*Each of the anomaly time series is computed relative to the mean of its respective model run.*"
4) NR anomalies are calculated with the respect to the mean of NR run.
5) NR is not the benchmark of OL or DA anomalies. OL anomaly is calculated with the respect to the mean of OL. DA anomalies follow the same rule.
6) We analyzed the anomalies because they are unitless and this is good practice when comparing the impact DA has on different variables. Nevertheless, we understand the value of showing actual values and, in the revised manuscript, we added these time series as Figure 3:

[Figure]

Figure 3. Global averaged daily values of LAI and five water variables (2011-06-01 to 2013-05-30).

Section 3.1 LAI

*"Figure 3a and Figure 4a show the time series of global averaged LAI values and corresponding anomalies, respectively. As expected, LAI values are largely impacted by the extreme precipitation conditions. The wet condition introduces more vegetation, while the dry condition limits the vegetation growth throughout the two-year period. The DA procedure effectively corrects the LAI errors caused by the biased precipitation input."*

Section 3.2 Water fluxes and storages

*"Daily time series of global averaged values and corresponding anomalies of the five water variables are shown in Figure 3(b-f) and Figure 4(b-f), respectively. The model well simulates the seasonality of all water fluxes/storages considered here. The OL runs reveal that global average values of all five variables are impacted by the highly biased precipitation conditions."*

- L. 243 ("thus the NCRMSE . . . becomes smaller"): Why?

In JJA, the stomatal closure can help to preserve water. So, the system does not lose too much water under the dry condition which result in smaller difference between DA-dry and the NR truth, and consequently shows smaller NCRMSE.

- L. 276-277 (LAI assimilation unable to correct for dry precipitation bias): Why?

A dry precipitation bias means that the system has (erroneously) has less water than in reality (NR in the synthetic experiment). Since no water is otherwise added to the system, LAI DA cannot fully correct water-related model states (such as soil moisture). The manuscript has been modified as below:

*"However, LAI assimilation is not able to correct the model when the input precipitation is negatively biased (dry condition). A dry precipitation bias means that the system has (erroneously) less water than in reality (NR in the synthetic experiment). Since no water is otherwise added to the system, LAI DA cannot fully correct water-related model states (such as soil moisture). The NCRMSEs of DA runs are either the same as in the OL runs (ET/CIE/CWS) or worse (SSM/TWS)."*

- L. 320-322: I don't see the logics. I would expect that large water-holding capacity would enhance the impact of LAI DA.

Our thought is that the LAI can affect soil moisture by changing the model's surface water condition. Over forest and woodland, the surface water condition is not changing much due to the large soil reservoir.

- L. 323 (forests and woodlands): Is this because of large rooting depth?

Large rooting depth is an important fact. Some discussion has been added to the manuscript: "*In other words, forest and woodland tend to have lower sensitivity in response to the change of precipitation conditions because of their large rooting depth.*"

- L. 331: Water-holding? Do you mean interception reservoir or soil reservoir?

It is soil reservoir. We changed it in the manuscript. "*This is due to large soil reservoir of forests and woodlands ……*"

- L. 374-375: This could be because the used DA system is not able to analysed RZSM from LAI observations. Please explain.

In the Noah-MP model, the relationship between LAI and soil moisture is very complex and indirect. So, the current LAI DA system is not able to have much of an effect on surface moisture at all depth.

**Editorial comments:**

- L. 251 (Figure 4): Color scale is difficult to interpret. Please use several colors (e.g. blue in addition to red).

We changed the color scale of Figure 4.

[Figure]

Figure 4. Maps of LAI NCRMSE for the OL and DA runs.

- L. 289 (Figure 5): Time axis labels are not readable. Please improve!
We enlarged the font size of the axis label and showed the time less frequently (every 3-month).
Please check below. We also enlarged the axis font size of all figures in the manuscript.

[Figure]

Figure 5. Monthly averaged NCRMSE for LAI and five water variables over the Northern hemisphere.

- L. 291 (Figure 6): Time axis labels are not readable. Please improve!
We enlarged the font size of the axis label and showed the time less frequently (every 3-month). Please check below.

[Figure]

Figure 6. Same as in Figure 5, but for the Southern hemisphere.

---

## Author Response (AR1)

The authors would like to thank the reviewer for their time, effort, and detailed comments. All suggestions were incorporated into the manuscript and explained in this response to reviewer document. We also thoroughly proofread and revised the whole manuscript.

**General comments**
The authors aim to assess to what extent the Noah-MP model can be optimized through the assimilation of leaf area index (LAI) observations at global scale. By utilizing two observing system simulation experiments (OSSEs) and the EnKF algorithm, the efficiency of assimilating LAI and model performance for water related variables are discussed. At first in my opinion this manuscript needs to be proofread/revised carefully for academic writing.
We would like to thank the reviewer. We have carefully proofread the revised manuscript.

Something that I do not understand is that the authors use the simulated LAI from the nature run as the 'truth' instead of observations. If nature run can achieve the "truth", why did the authors conduct assimilation based on different conditions (wet or dry)?
We chose to use an Observing System Simulation Experiment (OSSE) to quantify the potential impact of LAI assimilation on water variables simulated by the Noah-MP model while the forcing precipitation is affected by severe biases.

The forcing precipitation is usually provided by either reanalysis or satellite products. Such products are often affected by large biases (and random errors), which consequently affect the accuracy of the modeled variables. The question we want to answer here is: *When the forcing precipitation is biased, is LAI data assimilation able to improve the model estimates?* A real case study would certainly be of interest but in-situ observations (taken as reference) would also be affected by uncertainty, making it difficult to draw meaningful conclusions regarding the methodology itself. The proposed OSSE should serve as a feasibility test to quantify the potential of the proposed framework.

In an OSSE, i) the nature run (NR) intends to mimic the true input (including *unbiased* precipitation), LAI, and all water variables, ii) the open-loop run (OL) adds biases to the forcing precipitation (i.e., double or half the original value) to mimic the error in the precipitation product which will also produce biased model outputs of LAI and water variables; iii) the data assimilation (DA) run applies LAI DA to the OL run. We named the model run with double precipitation as wet

condition, and named the run with half precipitation as dry condition to describe the wet/dry bias that these two runs represent.

We modified the manuscript to clarify the OSSE design in section 2.2:

*"First, the Noah-MP model is spun-up for a 10-year period (2001-2010) to ensure a physically realistic state of equilibrium. Second, the model is run for a 29-month period (January 2011 – May 2013) to conduct the Nature Run (NR) with the same configuration as the spin-up one. By definition, an OSSE is a controlled experiment that does not assimilate any real observation. Instead, it treats all the model outputs from the NR as the "true" condition (denoted as the "synthetic truth"). The "true" LAI (i.e., the LAI output from NR) is then perturbed via a simple additive error model to produce the synthetic observations to be assimilated into the DA runs. The spin-up run and NR are forced by the original MERRA-2 precipitation data. Third, two Open Loop (OL) runs (no DA) are conducted for the same 29-month period under two conditions: i) "extremely dry" condition (the model is forced by halving the MERRA-2 precipitation data; OL-dry), and ii) "extremely wet" condition (the model is forced by doubling the MERRA-2 precipitation; OL-wet). The biased forcing precipitation data in OL mimic typical precipitation biases in current precipitation reanalysis and satellite products (e.g., Ghatak et al. 2018; Yoon et al. 2019).The two DA runs are then conducted under the two same conditions (DA-dry and DA-wet) using a one-dimensional EnKF assimilation algorithm, which is a built-in DA method in LIS ……"*

Other important comment is that why did the authors use the precipitation which are extremely biased instead of using a more precise precipitation forcing. Furthermore, did the authors run the assimilation experiment using the MERRA-2 precipitation instead of halving or doubling the value?

The 10-year spin-up run and the nature run are forced by the original MERRA-2 precipitation data. The OL and DA runs are forced by a perturbed (i.e., biased) version of the MERRA-2 precipitation. As described above, the OSSE uses this input in the OL and DA runs to mimic common biases in currently available precipitation products.

In conclusion, the manuscript in its current form suffers from several issues that prevent it to be published as is. In my opinion the paper still worth to be published after addressing all these issues, and a major revision is asked.

**Specific comments**

1. P3L56-57: As far as I know, LSMs not only couple with dynamic vegetation models, but also involve some dynamic vegetation modules. So the statement is not appropriate.

We changed "*dynamic vegetation model*" to "*dynamic vegetation module*" in the manuscript.

2. Section 2.2: Why do you use the precipitation forcing data which are strongly biased.

In the OSSE study, we use biased precipitation data in OL and DA runs to mimic precipitation biases that are very common in current precipitation reanalysis and satellite products (e.g., Ghatak et al. 2018, Yoon et al. 2019. These two references have been added to text in section 2.2:

*Ghatak, D., Zaitchik, B., Kumar, S., Matin, M. A., Bajracharya, B., Hain, C., & Anderson, M. (2018). Influence of Precipitation Forcing Uncertainty on Hydrological Simulations with the NASA South Asia Land Data Assimilation System. Hydrology, 5(4), 57. https://doi.org/10.3390/hydrology5040057*

*Yoon, Y., Kumar, S. V., Forman, B. A., Zaitchik, B. F., Kwon, Y., Qian, Y., Rupper, S., Maggioni, V., Houser, P., Kirschbaum, D., Richey, A., Arendt, A., Mocko, D., Jacob, J., Bhanja, S., & Mukherjee, A. (2019). Evaluating the Uncertainty of Terrestrial Water Budget Components Over High Mountain Asia. Frontiers in Earth Science, 7. https://doi.org/10.3389/feart.2019.00120*

3. Why did you choose the LAI simulations from the nature run as the "truth" instead of using the LAI observations? As you have described the reasons from P9L171 to L172, there are many other LAI products without missing data which can be used for assimilation.

By definition, an OSSE is a controlled experiment that does not assimilate any real observation. Instead, it treats all the model output from the nature run as the "true" condition. The LAI from the nature run is also considered as the true. We then perturbed it with a simple additive error model to produce synthetic observations to be assimilated into the model (DA run). Some explanation was added in section 2.2:

*"Second, the model is run for a 29-month period (January 2011 – May 2013) to conduct the Nature Run (NR) with the same configuration as the spin-up one. By definition, an OSSE is a controlled experiment that does not assimilate any real observation. Instead, it treats all the model outputs from the NR as the "true" condition (denoted as the "synthetic truth"). The "true" LAI (i.e., the LAI output from NR) is then perturbed via a simple additive error model to produce the synthetic observations to be assimilated into the DA runs. The spin-up run and NR are forced by the original MERRA-2 precipitation data."*

4. Did you evaluate the LAI or other variables from the natural run by using remote sensing LAI datasets or other kinds of observations?

As mentioned above, The LAI from the nature run is considered as the truth in the OSSE framework. The same LAI is perturbed with a simple additive error model to produce synthetic observations of LAI that are assimilated in the DA experiment. The LAI from OL or DA run is evaluated against the synthetic LAI observation from the nature run.

5. P9L178-P9L184: How did you determine the values of multiplicative perturbations (such as, the shortwave radiation and precipitation with a mean of 1 and standard deviations of 0.3 and 0.5, the standard deviation for longwave radiation of 50 W/m2, the standard deviation for LAI of 0.1)?

The forcing data perturbation applied here used the same perturbations as found in the literature below. We mentioned these past studies in section 2.2 "*Similar to previous work (Kumar et al. 2014, 2019a, 2019b), the MERRA-2 forcing inputs such as shortwave/longwave radiations and precipitation are perturbed hourly……*".

*Kumar, S. V., Peters-Lidard, C. D., Mocko, D., Reichle, R., Liu, Y., Arsenault, K. R., Xia, Y., Ek, M., Riggs, G., Livneh, B. and Cosh, M.: Assimilation of remotely sensed soil moisture and snow depth retrievals for drought estimation, J. Hydrometeorol., 15, 2446-2469, https://doi.org/10.1175/JHM-D-13-0132.1, 2014.*

*Kumar, S. V., Jasinski, M., Mocko, D. M., Rodell, M., Borak, J., Li, B., Beaudoing, H. K. and Peters-Lidard, C. D.: NCA-LDAS land analysis: Development and performance of a multisensor, multivariate land data assimilation system*

*for the National Climate Assessment, J. Hydrometeorol., 20, 1571-1593, https://doi.org/10.1175/JHM-D-17-0125.1, 2019.*

*Kumar, S. V., Mocko, D. M., Wang, S., Peters-Lidard, C. D. and Borak, J.: Assimilation of remotely sensed Leaf Area Index into the Noah-MP land surface model: Impacts on water and carbon fluxes and states over the Continental US, J. Hydrometeorol., 20, 1359-1377, https://doi.org/10.1175/JHM-D-18-0237.1, 2019.*

6. Have the evaluation and error metrics been used in former studies? If so, please list at least one references.

The equation for the Normalized Information Contribution (NIC) index is similar to the NIC used by Kumar et al. 2016. We added this reference to the text:

*Kumar, S.V., Zaitchik, B.F., Peters-Lidard, C.D., Rodell, M., Reichle, R., Li, B., Jasinski, M., Mocko, D., Getirana, A., De Lannoy, G. and Cosh, M.H.: Assimilation of gridded GRACE terrestrial water storage estimates in the North American Land Data Assimilation System, J. Hydrometeorol., 17, 1951-1972, https://doi.org/10.1175/JHM-D-15-0157.1, 2016*

7. How did you determine the initial conditions?

Initial conditions are obtained by a 10-year spin-up run. The spin-up run is described in the second paragraph of section 2.2.

8. The discussion section should include the discussion of the results in the context of other papers dealing with the same of similar subjects.

We added some discussion of past work on similar subjects:

*"Overall the improvement of water variables through LAI assimilation is not remarkable enough to compensate the model degradation caused by the biased precipitation forcing data. Previous studies (Pauwels et al. 2007; Sabater et al. 2007; Barbu et al. 2011; Fairbairn et al. 2017; Albergel et al. 2017) have tested the performance of the joint assimilation of LAI and soil moisture over regional domains and showed promising results. However, no experiment was performed at the global scale. Future work could investigate a multi-variate data assimilation system that concurrently merges both LAI and soil moisture (or TWS) observations globally."*

9. A more in-depth analysis of the results is necessary. In this paper the authors only talk about the statistical characteristic variables (such as the NCRMSE, NIC, etc) of LAI and water related variables. Why not focus on the LAI and water related variables themselves?

Sections 3.1 and 3.2 have been modified to add more analyses. Time series of global averaged LAI and water variables (Figure 3) were also added to the manuscript to provide more information on the actual variables (rather than anomalies). The discussion section (3.3) was also modified to provide more in-depth interpretation of the results.

[Figure]

Figure 3. Global averaged daily values of LAI and five water variables (2011-06-01 to 2013-05-30).

10. Why only perturb the meteorological forcing and not the initial conditions and/or model parameters?

We perturbed precipitation and radiation forcings because deemed dominant in water variable simulated by land surface models. Perturbing initial condition and model parameters is certainly an option that could be investigated in future studies. This recommendation has been added to the conclusion section.

"*Future research should focus on alternative DA methods, such as updating other related model states while assimilating LAI observations, perturbing the model initial condition and model*

*parameters, and/or assimilating actual satellite-based LAI observations (e.g., MODIS, GLASS) at the global scale to verify the efficiency of the proposed vegetation DA framework."*

11. How sensitive is LAI with respect to the meteorological forcing?

LAI is very sensitive to the forcing precipitation data. The wet and dry conditions have large impacts on the magnitude of LAI. The revised manuscript shows the time series of LAI values and anomalies. Below are the figures and description we added:

Section 3.1 LAI

*"Figure 3a and Figure 4a show time series of global averaged LAI values and corresponding anomalies, respectively. As expected, LAI values are largely impacted by the extreme precipitation conditions. The wet condition introduces more vegetation, while the dry condition limits the vegetation growth throughout the two-year period. The DA procedure effectively corrects the LAI errors caused by the biased precipitation input."*

[Figure]

*Figure 3a (top) and Figure 4a (bottom): Global averaged daily values of LAI and LAI anomalies*

**Technical corrections**

1. P2L27-L28: Can you illustrate which land surface model you use here? And the same to P2L38, P5L104, and so on.

We added "Noah-MP" to these three sentences.

2. P2L28-L29: Remove "the" from the phrase of "at the global scale", and the same to P5L100, P5L100, P22L361, and so on.

"the" was removed from "at the global scale" phrases.

3. P3L44: Do not need to leave two blank spaces here.

This has been fixed.

4. P3L46: It's not appropriate to use "between" among vegetation, precipitation, and soil moisture.

"between" was changed to "among".

5. P3L51: The related references cited here are not enough to illustrate the phenomenon that "these land surface processes and feedbacks have been examined through numerical modeling experiments". List more ......

More references have been listed: "*Foley et al. 1996; Kim and Wang 2007; Druel et al. 2019*"

6. P3L54: You needn't capitalizes the first letter for leaf area index.

This was fixed.

7. P4L67: "the Moderate Resolution Imaging Spectroradiometer" has been abbreviated to "MODIS" before.

This was removed.

8. P4L88-P5L90: Please refine this sentence.

The sentence was rewritten as "*Some water budget variables were improved through the assimilation procedure. The improvement is remarkable in agricultural areas because the assimilation added harvesting information to the model.*"

9. P5L95: Change "model simulated LAI" to "simulated LAI".

Removed.

10. P5L97: Please refine the statement of "focused on small regions".

Changed to "*…… and most of them are small region studies*".

11. P5L106-L107: Please define the abbreviation of all the water related variables when they first appear in this manuscript. Furthermore, "evapotranspiration" has been abbreviated to "ET" in P5L93.

The revised manuscript defined the abbreviation of all the water variables when they first appeared.

12. P5L110: Please specify which land surface model.
The title of section 2.1 was changed to "2.1. Land surface model (Noah-MP)".

13. P6L116-120: Please refine this sentence as it is too long.
The long sentence was divided into two sentences. "*Specifically, the prognostic vegetation growth combines a Ball-Berry photosynthesis-based stomatal resistance (Ball et al. 1987) with a dynamic vegetation model (Dickinson et al. 1998). The dynamic vegetation model calculates the carbon storages in various parts of the vegetation (leaf, stem, wood, and root) and the soil carbon pools.*"

14. P6L121: Please define "NASA".
Defined NASA as National Aeronautics and Space Administration.

15. P6L126: Keep the tense consistent.
Changed to "*The …….. (MERRA-2 ……) dataset serves as the meteorological forcings for Noah-MP.*"

16. P6L133-P7L138: Please define the abbreviation of all the water related variables when they first appear in this manuscript.
The revised manuscript defined the abbreviation of all the water variables when they first appeared.

17. P7L150: I am not sure whether the state of "a LAI EnKF" is appropriate.
Changed to "*the EnKF LAI assimilation*".

18. P7L153: The phase of "on a global scale" is not appropriate.
Changed to "*The proposed framework is evaluated through a global experiment (Antarctica excluded) at the 0.625° × 0.5° spatial resolution of the MERRA-2 forcing dataset (Figure 1).*"

19. P10L188-L189: Keep the tense consistent.
The sentence was changed to "*Thus, all the DA simulations are run for 20 members.*"

20. P10L194-L195: The water related variables have been defined before, and you can use their acronyms.
The variable names were changed to their acronyms.

21. P10L203: What does i and N in Equation 1 mean?
This explanation was added to the manuscript: "*N is the total number of X values, and i represents the index of each X value.*"

22. P10L208: the word "O" in the denominator looks like "zero" in Equation 2.
The letter "*O*" in "*OL*" does look similar to "zero", though "zero" is thinner. Hope the readers won't get confused.

$$C = \frac{E_{DA} - E_{OL}}{0 - E_{OL}}$$

23. P11L209: There are two periods.
Removed.

24. P12L220-L222: As Figure 3 shows the GLOBAL averaged LAI anomalies, it is better to use the statement of month (or JJA and SON seasons) instead of winter/summer season.
Changed to "*Moreover, the seasonality of LAI anomalies is evident, showing larger variations in DJF and JJA than during the transition periods (MAM and SON).*"

25. P12L229: Please refine this sentence.
The sentence was rewritten as "*Moreover, the DA runs show lower NCRMSEs than the corresponding OL runs across the globe (Figure 4) especially over shrublands and grasslands (refer to Figure 1 for land covers).*"

26. P12L241: Remove "the". Furthermore, this sentence is a little too long in my opinion.
Changed "*In the summer*" to "*In JJA*".
The long sentence was divided into two short ones. "*In JJA, the vegetation leaves in the north hemisphere are fully developed and the plants can use stomatal closure to preserve water under water limited condition (dry condition). Thus, the NCRMSE of dry condition becomes smaller and does not show much difference from the wet condition.*"

27. P14L263: Please change the "has higher chance" into "is more likely to".
Modified.

28. P14L268-269: I think this is the first appearance that positively biased is wet condition (or negatively biased is dry condition), or maybe earlier, and this statement does not need to be repeated each time it appears in this paper (see P14L277, P16L296-L297, P21L337, P21L339).
Most of the repetitions were removed. We kept the "wet/dry" in the conclusion section in case some readers check the conclusion before going through the whole manuscript.

29. P15L282-L287: It is better to use the statement of month instead of season.
All the season names in the manuscript were substituted by month.

30. Please add the description for the Y-coordinate for Figure 7, 8 and 9.
The Y-axis titles are added to Figure 7, 8 and 9 in the revised manuscript.

31. P21L357: Please specify which land surface model.

This was added to manuscript: *"This study evaluates the efficiency of assimilating vegetation information (i.e., LAI synthetic observations) within a land surface model (Noah-MP 3.6) when the precipitation forcing data are strongly biased (either positively or negatively)."*

Hydrol. Earth Syst. Sci. Discuss.,
https://doi.org/10.5194/hess-2019-504-RC2, 2019

[Figure]
The authors would like to thank the reviewer for their time, effort, and detailed comments. All suggestions have been incorporated into the manuscript and explained in this document. We also thoroughly proofread and revised the whole manuscript.

**General comments**
Synthetic observations are used to assess the impact of assimilating satellite-derived LAI estimates into the Noah land surface model. A major shortcoming of the assimilation system used in this study is that LAI assimilation has no direct impact on soil moisture. As a result, dry precipitation biases cannot be compensated for. This issue was at least partly solved in other assimilation systems. Unfortunately, the relevant literature is not completely cited. This paper is not well written, not complete for understanding, and cannot be published in the present form. Methods description is incomplete. Interpretation of results is made in the Result section instead of the Discussion section.

In the dry condition simulation, the amount of vegetation is less than in the reference simulation (what we call synthetic truth), due to a decrease (or even lack) in precipitation. When assimilating observations of LAI, we introduce more vegetation into the model, bringing it closer to the synthetic truth and consequentially improving CIE (canopy interception evaporation) and CWS (canopy water storage), which are directly related to LAI. However, variables that are not directly impacted by LAI, such as SSM (surface soil moisture), can hardly be improved by LAI assimilation solely. The poor performance of SSM in the dry condition experiment is mainly attributed to the fact that the amount of water in the model is conservative. Specifically, LAI assimilation introduces more vegetation, which requires more water than what available in the system (i.e., soil). Past work attempted to solve this problem by jointly assimilating LAI and soil moisture (Pauwels et al. 2007; Sabater et al. 2007; Barbu et al. 2011; Fairbairn et al. 2017; Albergel et al. 2017). We added some discussion on this topic and cited all these articles in discussion section of the revised manuscript (more detail is provided in our response to the Reviewer's specific comments below).

*Albergel, C., Munier, S., Leroux, D. J., Dewaele, H., Fairbairn, D., Barbu, A. L., ... & Le Moigne, P. (2017). Sequential assimilation of satellite-derived vegetation and soil moisture products using SURFEX_v8. 0: LDAS-Monde assessment over the Euro-Mediterranean area. Geoscientific Model Development, 10(10), 3889-3912.*

*Barbu, A. L., Calvet, J. C., Mahfouf, J. F., Albergel, C., & Lafont, S. (2011). Assimilation of Soil Wetness Index and Leaf Area Index into the ISBA-A-gs land surface model: grassland case study. Biogeosciences, 8(7), 1971-1986.*

*Fairbairn, D., Barbu, A., Napoly, A., Albergel, C., Mahfouf, J. F., & Calvet, J. C. (2017). The effect of satellite-derived surface soil moisture and leaf area index land data assimilation on streamflow simulations over France. Hydrology and Earth System Sciences, 21(4), 2015-2033.*

*Pauwels, V. R., Verhoest, N. E., De Lannoy, G. J., Guissard, V., Lucau, C., & Defourny, P. (2007). Optimization of a coupled hydrology–crop growth model through the assimilation of observed soil moisture and leaf area index values using an ensemble Kalman filter. Water Resources Research, 43(4).*

*Sabater, J. M., Rüdiger, C., Calvet, J. C., Fritz, N., Jarlan, L., & Kerr, Y. (2008). Joint assimilation of surface soil moisture and LAI observations into a land surface model. Agricultural and forest meteorology, 148(8-9), 1362-1373.*

**Recommendation: major revision.**

**Particular comments:**
- L. 39-40: Examples of joint assimilation of LAI and soil moisture in a land surface model can be found in the literature.

We have reviewed several studies that used LAI-soil moisture joint assimilation (Pauwels et al. 2007; Sabater et al. 2007; Barbu et al. 2011; Fairbairn et al. 2017; Albergel et al. 2017) and cited them in the discussion section:

*"Overall the improvement of water variables through LAI assimilation is not remarkable enough to compensate the model degradation caused by the biased precipitation forcing data. Previous studies (Pauwels et al. 2007; Sabater et al. 2007; Barbu et al. 2011; Fairbairn et al. 2017; Albergel et al. 2017) have tested the performance of the joint assimilation of LAI and soil moisture over regional domains and showed promising results."*

All the cited LAI-SM joint DA studies were conducted over regional domains. We emphasized our study is "at global scale" in the end of the discussion section to make the statement more accurate: *"However, no experiment was performed at the global scale. Future work could investigate a multi-variate data assimilation system that concurrently merges both LAI and soil moisture (or TWS) observations globally."*

- L. 95-97: In the same context and at the continental scale, Albergel et al. showed that sequential LAI assimilation can be used to analyse soil moisture at various depth, in addition to vegetation biomass (https://doi.org/10.5194/gmd-10-3889-2017). This property is particularly useful in dry conditions, when surface soil moisture tends to be decoupled from deeper soil layers.

Thank you for pointing us to this reference. We added it to the manuscript:

*"Only a few studies discussed the influences of LAI assimilation on the estimation of water variables such as soil moisture or streamflow (Pauwels et al. 2007; Sabater et al. 2008) and most of them focused on limited regions. Most recently, Albergel et al. (2017) conducted a study on a much larger domain – Europe and the Mediterranean basin –and showed improvement in soil moisture at various depths thanks to LAI assimilation."*

We also added this study to the discussion section as shown in the answer above.

*Albergel, C., Munier, S., Leroux, D. J., Dewaele, H., Fairbairn, D., Barbu, A. L., ... & Le Moigne, P. (2017). Sequential assimilation of satellite-derived vegetation and soil moisture products using SURFEX_v8. 0: LDAS-Monde assessment over the Euro-Mediterranean area. Geoscientific Model Development, 10(10), 3889-3912.*

- L. 109 (Section 2): A section describing the DA method is needed. What are the analysed variables? Does LAI DA impacts soil moisture?

The DA method in this study is implemented within the NASA Land Information System (LIS). The method has been applied in many sequential data assimilation studies. We added brief description and references for the DA method in section 2.2:

*"The two DA runs are then conducted under the two same conditions (DA-dry and DA-wet) using a one-dimensional EnKF assimilation algorithm, which is a built-in DA method in LIS. The EnKF DA algorithm is suitable for non-linear and intermittent land surface processes (Reichle et al. 2002a, 2002b). Details of the algorithm can be found in numerous previous studies (Reichle et al. 2010; De Lannoy et al. 2012; Liu et al. 2015; Kumar et al. 2019a)."*

- L. 138: How are subsurface waters represented? Do you represent inundations plains? Lakes?

TWS is the sum of snow water equivalent, surface water, soil moisture, and groundwater. So, subsurface water (i.e., groundwater) is included. Lakes and inundation plains are considered as surface water, which is also included in TWS. This information was added to Section 2.1 as follows:

*"…and TWS (defined as the sum of all water storage on the land surface and in the subsurface, including snow water equivalent, surface water, soil moisture, and groundwater [mm])."*

- L. 188 (ensemble members): How is this ensemble generated?

The model ensemble is generated by perturbing the meteorological forcing inputs (precipitation and shortwave/longwave radiations). Section 2.2 discusses all the details:

*"Similar to previous work (Kumar et al. 2014, 2019a, 2019b), the MERRA-2 forcing inputs such as shortwave/longwave radiations and precipitation are perturbed hourly. Multiplicative perturbations are applied to the shortwave radiation and precipitation with a mean of 1 and standard deviations of 0.3 and 0.5, respectively. The longwave radiation is perturbed via an additive perturbation with a standard deviation of 50 W/m2. The perturbations of the three meteorological forcing variables also include cross correlations: cross correlation between shortwave radiation and precipitation is -0.8, cross correlation between longwave radiation and precipitation is 0.5; and cross correlation between shortwave and longwave radiations is -0.5."*

*Kumar, S. V., Peters-Lidard, C. D., Mocko, D., Reichle, R., Liu, Y., Arsenault, K. R., Xia, Y., Ek, M., Riggs, G., Livneh, B. and Cosh, M.: Assimilation of remotely sensed soil moisture and snow depth retrievals for drought estimation, J. Hydrometeorol., 15, 2446-2469, https://doi.org/10.1175/JHM-D-13-0132.1, 2014.*

*Kumar, S. V., Jasinski, M., Mocko, D. M., Rodell, M., Borak, J., Li, B., Beaudoing, H. K. and Peters-Lidard, C. D.: NCA-LDAS land analysis: Development and performance of a multisensor, multivariate land data assimilation system for the National Climate Assessment, J. Hydrometeorol., 20, 1571-1593, https://doi.org/10.1175/JHM-D-17-0125.1, 2019.*

*Kumar, S. V., Mocko, D. M., Wang, S., Peters-Lidard, C. D. and Borak, J.: Assimilation of remotely sensed Leaf Area Index into the Noah-MP land surface model: Impacts on water and carbon fluxes and states over the Continental US, J. Hydrometeorol., 20, 1359-1377, https://doi.org/10.1175/JHM-D-18-0237.1, 2019.*

Moreover, an ensemble size sensitivity test was conducted to choose the number of ensemble members needed in this study (please refer to the Figure below). We omitted this figure in the manuscript, but mentioned the sensitivity test in section 2.2:

*"To select the optimal ensemble size, a sensitivity test is performed for ensemble sizes spanning from 2 to 24 members. The number of ensemble members has a strong impact on the model results at small sizes, while the model performance tends to become steady when more than 20 ensemble members are considered. Thus, all the DA simulations are run for 20 members."*

[Figure]

- L. 196-197: Why are these instabilities generated by DA?

The initial condition for the OL and DA runs is generated by a 10-year spin-up run that uses the original MERRA-2 precipitation. The OL and DA runs are forced by either doubled or halved precipitation that is not consistent with the spin-up run. So, the model needs to run for a certain time before stabilizing. The figure below shows the global averaged LAI time series from the beginning of the simulation (Jan. 1st, 2011) to Dec. 31st, 2011. The LAI simulated by OL and DA runs does not get stable until around May. Therefore, we decided to eliminate the first 5-month model outputs in the analyses. We added this explanation in the manuscript in section 2.3.

*"The initial condition for the OL and DA runs is generated by a spin-up run that uses the original MERRA-2 precipitation as input. However, the OL and DA runs are forced by either doubled or halved precipitation, which is not consistent with the spin-up run and the model needs some time to stabilize. The first 5-month model outputs are therefore eliminated from the evaluation to avoid the model systematic instability at the beginning of the OL and DA simulations and the evaluation, thus, focused only on model outputs from 2011-06-01 to 2013-05-31."*

[Figure]

- L. 203 (Eq. 1): Why do you use NCRMSE and not standard score metrics such as RMSE or ubRMSE (i.e. standard deviation of differences)?
In the manuscript, we compared the result of LAI and five water related variables (ET, CIE, CWS, SSM, and TWS). Units of these variables are very different, which is why we decided to adopt unitless statistical metrics. UbRMSE is certainly another valid option.

- L. 217 (Figure 3): Please change evaporation units. Since these time series are daily, should be per day instead of per second. It seems that CWS anomalies are 3 order of magnitude larger than ET anomalies. Why? Define here what you mean by "anomaly" (not defined in the text). NR anomalies: with respect to what? Is NR the benchmark or not? Real values have to be showed at some stage. Not only anomalies.
The ET value shown in the figure refers to the model output, which is an average over the day with the unit "kg m-2 s-1". CWS is the canopy water storage which include the water stored in the leaves and the intercepted water. So, it is much larger than the ET.

We analyzed anomalies (rather than actual values) because they are unitless and this is good practice when comparing the impact DA has on different variables. The anomalies are defined in section 2.3. "Each of the anomaly time series is computed relative to the mean of its respective model run." The NR anomalies are calculated with the respect to the mean of NR run. OL anomaly is calculated with the respect to the mean of OL. DA anomalies follow the same rule.

Nevertheless, we understand the value of showing actual values and, in the revised manuscript, we added these time series as Figure 3. Please check below for the figures of variable actual values and anomalies and the related descriptions we added in the revised manuscript.

[Figure]

*Figure 3. Global averaged daily values of LAI and five water variables (2011-06-01 to 2013-05-30).*

[Figure]

*Figure 4. Global averaged daily anomalies of LAI and five water variables (2011-06-01 to 2013-05-30).*

Section 3.1 LAI
*"Figure 3a and Figure 4a show time series of global averaged LAI values and corresponding anomalies, respectively. As expected, LAI values are largely impacted by the extreme precipitation conditions. The wet condition introduces more vegetation, while the dry condition limits the vegetation growth throughout the two-year period. The DA procedure effectively corrects the LAI errors caused by the biased precipitation input."*

Section 3.2 Water fluxes and storages
*"Daily time series of global averaged values and corresponding anomalies of the five water variables are shown in Figure 3(b-f) and Figure 4(b-f), respectively. The model well simulates the seasonality of all water fluxes/storages considered here. The OL runs reveal that global average values of all five variables are impacted by the highly biased precipitation conditions."*

- L. 243 ("thus the NCRMSE . . . becomes smaller"): Why?
In JJA, the stomatal closure can help to preserve water. So, the system does not lose too much water under the dry condition which result in smaller difference between DA-dry and the NR truth, and consequently shows smaller NCRMSE.

- L. 276-277 (LAI assimilation unable to correct for dry precipitation bias): Why?
A dry precipitation bias means that the system has (erroneously) has less water than in reality (NR in the synthetic experiment). Since no water is otherwise added to the system, LAI DA cannot fully correct water-related model states (such as soil moisture). The manuscript has been modified as below:
*"However, LAI assimilation is not able to correct the model when the input precipitation is negatively biased (dry condition). A dry precipitation bias means that the system has (erroneously) less water than in reality (NR in the synthetic experiment). Since no water is otherwise added to the system, LAI DA cannot fully correct water-related model states (such as soil moisture). The NCRMSEs of DA runs are either the same as in the OL runs (ET/CIE/CWS) or worse (SSM/TWS)."*

- L. 320-322: I don't see the logics. I would expect that large water-holding capacity would enhance the impact of LAI DA.
Our thought is that the LAI can affect soil moisture by changing the model's surface water condition. Over forest and woodland, the surface water condition is not changing much due to the large soil reservoir.

- L. 323 (forests and woodlands): Is this because of large rooting depth?
Large rooting depth is an important fact. Some discussion has been added to the manuscript:
*"In other words, forest and woodland tend to have lower sensitivity in response to the change of precipitation conditions because of their large rooting depth."*

- L. 331: Water-holding? Do you mean interception reservoir or soil reservoir?
It is soil reservoir. We changed it in the manuscript:
*"This is due to large soil reservoir of forests and woodlands ……"*

- L. 374-375: This could be because the used DA system is not able to analysed RZSM from LAI observations. Please explain.
In the Noah-MP model, the relationship between LAI and soil moisture is very complex and indirect. So, the current LAI DA system is not able to have much of an effect on surface moisture at all depth.

**Editorial comments:**

- L. 251 (Figure 4): Color scale is difficult to interpret. Please use several colors (e.g. blue in addition to red).

We changed the color scale of Figure 4.

[Figure]

*Figure 5. Maps of LAI NCRMSE for the OL and DA runs.*

- L. 289 (Figure 5): Time axis labels are not readable. Please improve!
We enlarged the font size of the axis label and showed the time less frequently (every 3-month).
Please check below. We also enlarged the axis font size of all figures in the manuscript.

[Figure]

*Figure 6. Monthly averaged NCRMSE for LAI and five water variables over the Northern hemisphere.*

- L. 291 (Figure 6): Time axis labels are not readable. Please improve!
We enlarged the font size of the axis label and showed the time less frequently (every 3-month).
Please check below.

[revised manuscript text omitted]

---

## Author Response (AR2)

Report was submitted on 30 April 2020

**General comments**
This is the second time I am reviewing this manuscript, and thanks to the authors for taking into account of my previous comments. Authors did a fair job addressing my reviews, and the new version of the manuscript has been significantly improved. However, I still have an issue with the relationship between LAI and terrestrial water fluxed and storages variables (ET, CIE, CWS, SSM, TWS), at least in the section of discussion. The authors would pay more attention in the connection of LAI with these variables, or is there a relationship among the variables themselves? The paper has potential for publication in HESS after revisions.

The authors would like to thank the reviewer for their time and effort. As suggested, we have revised the Discussion section by adding a paragraph that investigates the relationships between LAI and water variables.

[revised manuscript text omitted]